# ADU-Depth: Attention-based Distillation with Uncertainty Modeling for Depth Estimation

**Zizhang Wu**[1,2†]     **Zhuozheng Li**[2†]     **Zhi-Gang Fan**[2]     **Yunzhe Wu** [2]

**Xiaoquan Wang** [3]     **Rui Tang** [2]     **Jian Pu**[1✉]

[1]**Fudan University,** [2]**ZongmuTech,** [3]**ExploAI**
wuzizhang87@gmail.com, {zhuozheng.li, zhigang.fan, nelson.wu, rui.tang}@zongmutech.com,
rocky.wang@exploai.com, jianpu@fudan.edu.cn [*]

**Abstract:** Monocular depth estimation is challenging due to its inherent ambiguity and ill-posed nature, yet it is quite important to many applications. While recent works achieve limited accuracy by designing increasingly complicated networks to extract features with limited spatial geometric cues from a single RGB image, we intend to introduce spatial cues by training a teacher network that leverages left-right image pairs as inputs and transferring the learned 3D geometry-aware knowledge to the monocular student network. Specifically, we present a novel knowledge distillation framework, named ADU-Depth, with the goal of leveraging the well-trained teacher network to guide the learning of the student network, thus boosting the precise depth estimation with the help of extra spatial scene information. To enable domain adaptation and ensure effective and smooth knowledge transfer from teacher to student, we apply both attention-adapted feature distillation and focal-depth-adapted response distillation in the training stage. In addition, we explicitly model the uncertainty of depth estimation to guide distillation in both feature space and result space to better produce 3D-aware knowledge from monocular observations and thus enhance the learning for hard-to-predict image regions. Our extensive experiments on the real depth estimation datasets KITTI and DrivingStereo demonstrate the effectiveness of the proposed method, which ranked 1st on the challenging KITTI online benchmark[2].

**Keywords:** Monocular depth estimation, Camera perception, Distillation

## 1 Introduction

Monocular depth estimation, with the goal of measuring the per-pixel distance from a single camera perception, is absolutely crucial to unlocking exciting robotic applications such as autonomous driving [1, 2], 3D scene understanding [3, 4] and augmented reality [5, 6]. However, it is quite difficult to obtain precise depth values from only a single 2D input, since monocular depth estimation is ill-posed and inherently ambiguous [7, 8] where many 3D scenes can actually give the same input picture [9, 10]. Early depth estimation methods relied primarily on hand-crafted features [11, 12], geometric assumptions [12], and non-parametric depth transfer [13] and achieved limited success.

With the rise of deep learning techniques, the performance of monocular depth estimation has seen a significant boost [14, 15, 16]. For supervised learning, a list of approaches focus on designing deeper and more complicated networks to improve depth prediction under the supervision of ground-truth depth [17, 18, 19]. The current trend has been to combine convolutional neural networks (CNNs) with vision transformers and attention mechanisms [20, 21, 22, 23], which can extract more meaningful features in an image and capture long-range dependencies. Inspired by the works of

---

[*]The denotion † means these authors contributed equally and ✉ means the corresponding author.
[2]From 24 Jun. 2022 to 22 Feb. 2023.

7th Conference on Robot Learning (CoRL 2023), Atlanta, USA.

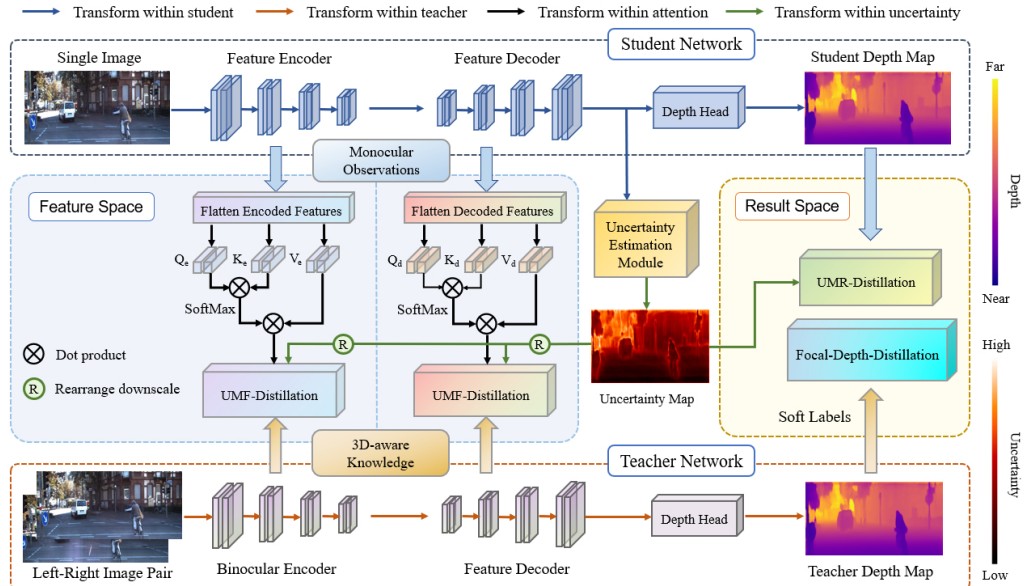

Figure 1: Illustration of ADU-Depth, which consists of a student network, a teacher network and the distillation scheme. We design an attention-adapted feature distillation and a focal-depth-adapted response distillation. Meanwhile, we introduce an uncertainty estimation module to guide the left-right knowledge transfer, which are UMF-Distillation and UMR-Distillation, respectively. During inference, only the monocular student is employed with only a single image as input.

Conditional Random Fields (CRFs) [24, 11, 25], a recent method [9] was proposed to split the input into windows and incorporate the neural window fully-connected CRFs with a multi-head attention mechanism [26] to better capture the geometric relationships between nodes in the graph. However, considering that a single image contains limited 3D scene geometric information, this renders the monocular depth estimation task a difficult fitting problem without extra depth cues [27, 10].

In this work, we present a novel distillation framework named ADU-Depth. We show that it is able to assist top depth predictors to infer more accurate depth maps, benefiting from 3D-aware knowledge transfer of left-right views as humans do. We first train a teacher network to learn 3D knowledge by concatenating the left-right images together and modify the model structure to accommodate the image pair inputs. This intuitive yet quite effective mechanism leads to a powerful teacher network and can be easily deployed for recent depth networks. Considering the strength of capturing geometric relationships between nodes in a graph, we use an encoder-decoder network similar to [9] as the student network and add an uncertainty branch. To ensure effective and smooth knowledge distillation, the proposed distillation framework should be capable of producing 3D-aware features and responses under the guidance of the teacher. Therefore, we introduce a self-attention adaptation module for feature imitation and shrink the feature domain gap. Next, we design a focal-depth-loss to more effectively distill high-quality responses from soft-labels provided by the teacher. Furthermore, we combine distillation with an uncertainty estimation module to learn the prediction distribution and better produce 3D-aware features and responses from monocular observations.

To the best of our knowledge, this is the first time in the supervised depth estimation community that left-right view knowledge is transferred from the teacher network to a monocular student. In summary, our contributions include: (i) a novel ADU-Depth framework that distills 3D-aware knowledge from a teacher network into a supervised monocular student; (ii) an attention adaptation module for both encoded and decoded feature distillation; (iii) a distillation-score based focal-depth-loss for response distillation and focuses on hard-to-transfer knowledge; and (iv) an uncertainty estimation module to guide the distillation in both feature and result spaces. The above techniques enable our method to achieve state-of-the-art performance of monocular depth estimation across all metrics on the KITTI and DrivingStereo datasets and ranked 1st on the KITTI online benchmark.

## 2  Related Work

**Monocular Depth Estimation.**   Neural network based methods have dominated most monocular depth estimation benchmarks. Eigen et al. [14, 15] first proposed a depth regression approach by utilizing two stacked CNNs to integrate global and local information. To model the ambiguous mapping between images and depth maps, Laina et al. [28] presented an end-to-end convolution architecture with residual learning. In contrast, Godard et al. [29] formulated depth estimation as a pixel-level classification task and also obtained depth prediction confidence. Besides, several multi-view methods [30, 2] employed a recurrent neural network based optimization technique that repeatedly alternates depth and pose updates to minimize the metric cost. Furthermore, many self-supervised monocular works took left-right image pairs [31, 32, 33, 34, 35, 36] or multi-frame images [37, 38, 39] as input for the network training. By benefiting from the vision transformers, recent supervised works [18, 21, 23, 22] employed the attention mechanism for more accurate depth estimation. More recently, the neural window FC-CRFs with multi-head attention was proposed to better capture the relationship between nodes in the graph and reduce the computational cost [9]. However, in the absence of 3D geometric information, these complicated transformer and attention based methods achieved limited performance gain for monocular depth estimation.

**Knowledge Distillation Methods.**   Knowledge distillation (KD) was first proposed in [40] to transfer the learned knowledge from a cumbersome teacher model to a light-weight student model through teacher-generated soft targets. Besides its success in image classification, KD strategy has been widely applied in objection detection [41, 42, 43], semantic segmentation [44], and depth estimation [45, 46, 47, 36, 48]. Pilzer et al. [46] claimed that distillation is particularly relevant to depth estimation due to the limitations of network size in practical real-time applications. They attempted to exploit distillation to transfer knowledge from the refinement teacher network to a self-supervised student. For fast depth estimation on mobile devices, Wang et al. [47] utilized knowledge distillation to transfer the knowledge of a stronger complex teacher network to a light-weight student network. More recently, DistDepth [48] distilled depth-domain structure knowledge into a self-supervised student to obtain accurate indoor depth maps. Unlike existing distillation methods for depth estimation, our ADU-Depth is the first to distill knowledge from left-right views into a supervised monocular depth predictor and incorporates novel distillation techniques into the framework.

**Uncertainty Estimation Methods.**   Uncertainty estimation plays a critical role in improving the robustness and performance of deep learning models [49, 50, 51, 52]. Particularly, it was also introduced into monocular depth estimation [53, 54, 55], which was mostly in self- and semi-supervised methods due to their high uncertainty. Poggi et al. [56] examined several manners of uncertainty modeling in monocular depth estimation and suggested the optimum strategy in various scenarios. Nie et al. [55] employed a student-teacher network to iteratively improve uncertainty performance and depth estimation accuracy through uncertainty-aware self-improvement. Inspired by the work [57] that introduces uncertainty estimation to better produce high-quality-like features from low-quality observations for degraded image recognition, we design an uncertainty estimation module and incorporate it into the proposed distillation framework for depth estimation, which can better produce 3D-aware features and responses from monocular observations.

## 3  Proposed Method

### 3.1  Overall Framework

As illustrated in Fig. 1, the framework of ADU-Depth can be separated into three parts (i.e., a teacher network, a monocular student, and the distillation scheme). We first pre-train the teacher network on left-right image pairs with dense ground-truth depth. We then freeze it to produce features and responses for training the student network on a single image by the proposed distillation strategy. The goal of our framework is to train the monocular student to learn to generate 3D-aware features and responses under the guidance of the left-right view based teacher network. To meet this goal,

we first introduce attention-adapted feature distillation and focal-depth-adapted response distillation to encourage feature and response imitation, respectively. Then, an uncertainty estimation module (UEM) is designed to model the uncertainty of both the feature and response distribution.

**Student model.** We employ an encoder-decoder network similar to the work [9] as our baseline depth predictor due to its promising performance and ability to capture relationships between the nodes in the graph. In detail, it adopts a multi-level encoder-decoder architecture, where four levels of neural window FC-CRFs modules are stacked. The model employs the swin-transformer [58] encoder to extract multi-level image features and the CRFs optimization decoder to predict the next-level depth according to the image features and coarse depth. We consider that the neural window FC-CRFs module and internal multi-head attention mechanism can better capture the critical scene geometry information of the left-right view data, hence outputting an optimized depth map. In particular, we further introduce an novel uncertainty estimation module alongside the depth head as another branch, which generates the per-pixel uncertainty to better guide the knowledge distillation.

**Teacher model.** To make it easier accessible and more flexible to deploy the teacher network and ensure the model training efficiency in practical applications, while reducing the structural domain gap between the teacher and student networks, our teacher model chooses to learn 3D knowledge by concatenating the left-view and right-view images together and modifies the initial three channels of the swin-transformer to six channels to accommodate the image pair inputs. This mechanism is simple yet quite effective and can avoid huge computations and domain gaps brought by complex structural changes. Furthermore, the concatenation operation is also used in the stereo version of the unsupervised method [32], where the additional 3D information of left-right images enables a more complete understanding of the input scene. Note that, left-right image pairs are commonly used for practical robotic applications, and many existing datasets are available [59, 60, 61].

### 3.2 Knowledge Distillation Strategy

In this section, we describe our three complementary distillation schemes: attention-adapted feature distillation, focal-depth-adapted response distillation, and uncertainty modeling based distillation.

**Attention-adapted feature distillation.** Due to the domain gap between features extracted from a single image and a left-right (stereo) image pair, we consider it sub-optimal to directly execute a monocular student to learn the feature representation of the teacher. While the 3D scene geometry information learned from the left-right data domain greatly enhances teacher performance, the effective transfer of learned knowledge from teacher to student is an obvious challenge. Since the number of input channels of the student is not compatible with those of the teacher, an adaptation module is important to align the former to the latter for calculating the distance metric and enabling an effective feature approximation between the student and teacher. Unlike the work in [43], which uses a full convolution adaptation layer, our experiments show that self-attention adaptation layers are more conducive to bridging the data domain gap and forcing the student to more effectively adapt and imitate the teacher's features. They are able to capture long-range feature dependencies and focus on the critical spatial cues of the image itself [26]. Specifically, the feature encoder extracts four levels of feature maps following Fig. 1. For each level of feature maps, we first flatten it and then employ a self-attention operation to obtain the attention-adapted feature $F_e^s$:

$$F_e^s = \mathrm{softmax}(Q_e^s \cdot K_e^s) \cdot X_e^s, \tag{1}$$

where $Q_e^s$, $K_e^s$, and $X_e^s$ are the learned query, key, and value of the student encoder, respectively. ($\cdot$) denotes dot production. Similarly, the attention-adapted decoded feature $F_d^s$ can also be obtained via Eq. (1), where $Q_d^s$, $K_d^s$, and $X_d^s$ are learned from the multi-level decoded features. Then, given the multi-level extracted feature maps of the teacher model $F_e^t$ and the attention-adapted encoded student feature $F_e^s$, the proposed encoded feature distillation loss $\mathcal{L}_e$ can be formulated as:

$$\mathcal{L}_e = \frac{1}{N} \left\| F_\mathbf{e}^\mathbf{t} - F_\mathbf{e}^\mathbf{s} \right\|_2^2. \tag{2}$$

Similarly, given the multi-level decoded feature maps of teacher model $F_d^t$ and the attention-adapted decoded student feature $F_d^s$, our decoded feature distillation loss $\mathcal{L}_d$ can be formulated as:

$$\mathcal{L}_d = \frac{1}{N} \left\| F_d^t - F_d^s \right\|_2^2. \tag{3}$$

**Focal-depth-adapted response distillation.** We use the predicted depth from the teacher as extra soft labels to transfer the learned knowledge to the student in the result space. Furthermore, inspired by the focal loss [62], we design a focal-depth loss to enhance the learning of hard-to-predict samples. Note that the focal loss is originally for classification and the $p \in [0, 1]$ in the focal loss is the estimated probability for the class with label $y = 1$. To enhance the student's learning of hard-to-transfer knowledge of depth estimation from the teacher network, we introduce a depth distillation score $p_d \in [0, 1]$ to indicate the degree of prediction approximation between the student and teacher:

$$p_d = 1 - \frac{1}{N} \sum_{i=1}^{N} \left| \frac{d_i^t - d_i^s}{d_i^t} \right|, \tag{4}$$

where $N$ is the number of pixels, $i$ is the pixel index. Then, we use the variant $\alpha_d$ to balance the importance of high distillation-score examples. The proposed focal-depth loss can be described as:

$$\mathcal{L}_{focal}(p_d) = -\alpha_d (1 - p_d)^\gamma \log(p_d), \tag{5}$$

where $\gamma \geq 0$ is a tunable focusing parameter, $d_i^t$ and $d_i^s$ denote the depth predictions of the teacher and student, respectively. Furthermore, we also use the soft label L1-loss for distillation in the result space and then combine it with uncertainty modeling, which is represented in Eq. (8). For each pixel $i$ in an image, the initial soft label response distillation loss is computed as:

$$\mathcal{L}_{rd} = \sum_{i=1}^{N} \left\| d_i^t - d_i^s \right\|_1. \tag{6}$$

**Uncertainty modeling based distillation.** To estimate the pixel-wise uncertainty, we design an uncertainty estimation module (UEM) as a new branch at the end of the backbone network. The UEM consists of a convolution layer, a sigmoid activation function, and a scaling layer. Based on the Bayesian deep learning framework [49], we train the depth network to predict the per-pixel log variance $s_i = \log \sigma_i^2$. The predicted uncertainty map has the same dimension as the depth map and is then rearranged to the size of multi-level feature maps, i.e. 1/4, 1/8, 1/16, and 1/32 of the original size. Given the Gaussian assumption for $\ell_2$ loss, the uncertainty modeling based feature distillation loss $\mathcal{L}_{umf}$ is defined as:

$$\mathcal{L}_{umf} = \frac{1}{N} \sum_{i=1}^{N} \left( \frac{1}{2} \exp(-s_i) \left\| F_i^t - g\left(\tilde{F}_i^s; s_i\right) \right\|_2^2 + \frac{1}{2} s_i \right), \tag{7}$$

where $F^t$ denotes left-right feature maps from the teacher network and $\tilde{F}^t = g\left(\tilde{F}_i^s; s_i\right)$ means the restored 3D-aware feature maps from the student network with estimated log variances $s_i$ through the uncertainty estimation module. For those $\tilde{F}^t$ far away from $F^t$, the network will predict larger variances to reduce the error term $\exp(-s_i) \left\| F_i^t - g\left(\tilde{F}_i^s; s_i\right) \right\|_2^2$, instead of overfitting to those erroneous regions. When $\tilde{F}_i^t$ is easy to learn, the second term $\frac{1}{2} s_i$ performs a major role in the loss function, and the network tends to make variances smaller. In this way, it works similarly to an attention mechanism, which enables the network to focus on the hard regions in the image or hard samples in the training set [63, 57]. $\mathcal{L}_{umf}$ is employed on both encoded and decoded features.

For the response distillation in result space, we choose Laplace uncertainty loss since it is more appropriate to model the variances of residuals with $\ell_1$ loss. Given the Laplace assumption, we use $\tilde{d}_i^t = g\left(\tilde{d}_i^s; s_i\right)$ to denote the learned stereo-aware student depth prediction. Then, the uncertainty modeling based response distillation (UMR-Distillation) loss $\mathcal{L}_{umr}$ is defined as:

$$\mathcal{L}_{umr} = \frac{1}{N} \sum_{i=1}^{N} \left( \sqrt{2} \exp(-s_i) \left\| d_i^t - g\left(\tilde{d}_i^s; s_i\right) \right\|_1 + s_i \right). \tag{8}$$

| Method | Reference | Sup | Distill | Sq Rel↓ | Abs Rel↓ | RMSE↓ | RMSE$_{log}$↓ | $\delta_1 < 1.25$↑ | $\delta_2 < 1.25^2$↑ | $\delta_3 < 1.25^3$↑ |
|---|---|---|---|---|---|---|---|---|---|---|
| Xu et al. [64] | CVPR 2018 | ✓ | ✗ | 0.897 | 0.122 | 4.677 | – | 0.818 | 0.954 | 0.985 |
| Guo et al. [45] | ECCV 2018 | ✓ | ✓ | 0.515 | 0.092 | 3.163 | 0.159 | 0.901 | 0.971 | 0.988 |
| Refine and Distill [46] | CVPR 2019 | ✗ | ✓ | 0.831 | 0.098 | 4.656 | 0.202 | 0.882 | 0.948 | 0.973 |
| Yin et al. [65] | ICCV 2019 | ✓ | ✗ | – | 0.072 | 3.258 | 0.117 | 0.938 | 0.990 | 0.998 |
| PackNet-SAN [66] | CVPR 2021 | ✓ | ✗ | – | 0.062 | 2.888 | – | 0.955 | – | – |
| PWA [20] | AAAI 2021 | ✓ | ✗ | 0.221 | 0.060 | 2.604 | 0.093 | 0.958 | 0.994 | **0.999** |
| DPT [21] | ICCV 2021 | ✓ | ✗ | – | 0.062 | 2.573 | 0.092 | 0.959 | 0.995 | **0.999** |
| SingleNet [36] | ICCV 2021 | ✗ | ✓ | 0.681 | 0.094 | 4.392 | 0.185 | 0.892 | 0.962 | 0.981 |
| AdaBins [18] | CVPR 2021 | ✓ | ✗ | 0.190 | 0.058 | 2.360 | 0.088 | 0.964 | 0.995 | **0.999** |
| NeWCRFs [9] | CVPR 2022 | ✓ | ✗ | 0.155 | 0.052 | 2.129 | 0.079 | 0.974 | **0.997** | **0.999** |
| P3Depth [10] | CVPR 2022 | ✓ | ✗ | 0.270 | 0.071 | 2.842 | 0.103 | 0.953 | 0.993 | 0.998 |
| Liu et al. [67] | TCSVT 2023 | ✗ | ✓ | 0.635 | 0.096 | 4.158 | 0.171 | 0.905 | 0.969 | 0.985 |
| ADU-Depth | – | ✓ | ✓ | **0.147** | **0.049** | **2.080** | **0.076** | **0.976** | **0.997** | **0.999** |

Table 1: Quantitative results on the KITTI Eigen split with a cap of 0-80m. Seven widely-used metrics are reported and calculated strictly following NeWCRFs [9]. The methods are divided by whether they are supervised (Sup) and the use of distillation (Distill).

| Method | Dataset | SILog↓ | Sq Rel↓ | Abs Rel↓ | iRMSE↓ | RMSE↓ | $\delta_1 < 1.25$↑ | $\delta_2 < 1.25^2$↑ | $\delta_3 < 1.25^3$↑ |
|---|---|---|---|---|---|---|---|---|---|
| DORN [17] | val | 12.22 | 3.03 | 11.78 | 11.68 | 3.80 | 0.913 | 0.985 | 0.995 |
| BA-Full [19] | val | 10.64 | 1.81 | 8.25 | 8.47 | 3.30 | 0.938 | 0.988 | 0.997 |
| NeWCRFs [9] | val | 8.31 | 0.89 | 5.54 | 6.34 | 2.55 | 0.968 | 0.995 | 0.998 |
| ADU-Depth | val | **6.64** | **0.61** | **4.61** | **5.25** | **1.98** | **0.981** | **0.997** | **0.999** |
| DORN [17] | online test | 11.77 | 2.23 | 8.78 | 12.98 | – | – | – | – |
| BA-Full [19] | online test | 11.61 | 2.29 | 9.38 | 12.23 | – | – | – | – |
| PackNet-SAN[66] | online test | 11.54 | 2.35 | 9.12 | 12.38 | – | – | – | – |
| PWA [20] | online test | 11.45 | 2.30 | 9.05 | 12.32 | – | – | – | – |
| NeWCRFs [9] | online test | 10.39 | 1.83 | 8.37 | 11.03 | – | – | – | – |
| ADU-Depth | online test | **9.69** | **1.69** | **7.26** | **9.61** | – | – | – | – |

Table 2: Quantitative results on the KITTI official split. Eight widely-used metrics are calculated for the validation set and four metrics from the KITTI official online server are used for the test set. Our method **ranked 1st** on the KITTI depth prediction benchmark (initially named "ZongDepth").

### 3.3 End-to-End Network Training

For the training of the teacher network, only the loss $L_b$ from the baseline method is adopted [9]. We train our student network in an end-to-end manner using the following loss function:

$$\mathcal{L} = \mathcal{L}_b + \lambda_1 \cdot \mathcal{L}_{umr} + \lambda_2 \cdot \mathcal{L}_{umf} + \lambda_3 \cdot \mathcal{L}_{focal}, \tag{9}$$

where $\lambda_1, \lambda_2, \lambda_3$ are the hyper-parameters to balance the loss of each module.

## 4 Experimental Results

### 4.1 Implementation Details

Our work is implemented in PyTorch and experimented on Nvidia RTX A6000 GPUs. The network is trained for 40 epochs with a mini-batch size of 4. The input image is resized as $352 \times 1120$ before being fed into the network and then rearranged and downscaled to four levels in the encoder-decoder network, i.e. $1/4, 1/8, 1/16, 1/32$. The learning rate is $2 \times 10^{-5}$. We set $\beta_1 = 0.9$ and $\beta_2 = 0.999$ in the Adam optimizer for network optimization. For KITTI dataset, $\lambda_1 = 0.9, \lambda_2 = 0.6, \lambda_3 = 0.8$. The output depth map is $1/4 \times 1/4$ size of the original image and finally resized to the full resolution. For the analysis of computation time, the inferred speed of our ADU-Depth is 3.82 FPS, which is faster than the 3.48 FPS of NeWCRFs [9] and without introducing an additional inference cost.

### 4.2 Evaluations

**Evaluation on KITTI.** We first evaluate our method on the KITTI Eigen split [14]. As shown in Table 1, our ADU-Depth achieves new state-of-the-art performance with significant improvements over other top performers and existing distillation-based depth estimation methods [45, 46, 36] with significant improvements. We then evaluate our method on the KITTI official split [59] as shown in Table 2. Obviously, our method achieves state-of-the-art performance across all the evaluation metrics compared to existing depth estimation approaches.

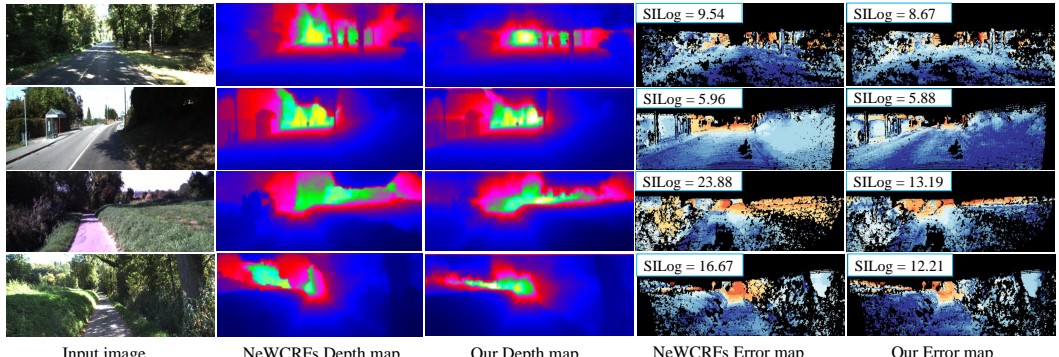

| Input image | NeWCRFs Depth map | Our Depth map | NeWCRFs Error map | Our Error map |

Figure 2: Qualitative results generated by the official server on the KITTI online benchmark.

| Method | Sq Rel ↓ | Abs Rel ↓ | RMSE ↓ |
|---|---|---|---|
| DORN [17] | 0.126 | 0.055 | 2.173 |
| Adabins [18] | 0.083 | 0.036 | 1.459 |
| NeWCRFs [9] | 0.071 | 0.032 | 1.310 |
| ADU-Depth | **0.064** | **0.028** | **1.146** |

Table 3: Quantitative results on the DrivingStereo.

| Settings | Sq Rel ↓ | Abs Rel ↓ | RMSE ↓ |
|---|---|---|---|
| Teacher | 0.074 | 0.040 | 1.418 |
| Baseline | 0.188 | 0.056 | 2.325 |
| - Focal | 0.152 | 0.050 | 2.103 |
| - UEM | 0.154 | 0.051 | 2.108 |
| - Attention | 0.161 | 0.052 | 2.127 |
| Full-Setting | 0.147 | 0.049 | 2.080 |

Table 4: Ablation study on the KITTI Eigen split.

**Evaluation on KITTI online test.** On the challenging KITTI online benchmark, our method ranked 1st among all submissions of depth prediction for seven months, while NewCRFs [9] ranked 12th. The performance gain is more obvious than its improvement on the Eigen split, since our method is better at predicting the depth of hard regions or test images with a domain gap. The scale-invariant logarithmic (SILog) error is the main ranking metric used to measure the relationships between points in the scene [14]. The lowest "SILog" error indicates that our method could better capture the relationships between nodes in the scene, reasoning as the learning of stereo-aware knowledge. We also provide qualitative comparison results in Fig. 2, including the predicted depth maps and the error maps. The error map depicts correct estimates in blue and wrong estimates in red color tones [68]. Dark regions denote the occluded pixels that fall outside the image boundaries. Our method predicts more accurate and reliable depths in various scenes, especially for hard-to-predict image regions, e.g., distant objects, repeated textures and weak light scenes. On the one hand, attention-adapted feature distillation allows our model to effectively learn more about the spatial information of the scene from the teacher network. On the other hand, the introduction of uncertainty and focal-depth made the model focus on the learning of difficult areas.

**Evaluation on DrivingStereo.** DrivingStereo [60] is a large-scale stereo dataset that consists over 180k images covering a diverse set of of driving scenarios. We further compare our method with several well-known methods on a subset of the DrivingStereo dataset with 7000 image pairs for training and 600 images only for testing. The quantitative results of our method compared with others are shown in Table 3, where four widely used evaluation metrics are calculated for the test set. Our ADU-Depth outperforms these monocular depth predictors in all evaluation metrics.

### 4.3 Qualitative results of Uncertainty Estimation

The uncertainty estimation module (UEM) should be capable of producing uncertainty values that are well aligned with the depth estimation errors. Fig. 3 shows several examples of our predicted uncertainty maps, where brighter colors indicate high uncertainty. We can see that the high uncertainty values are distributed on the distant background region and the object boundaries. These uncertainty values are quite reasonable according to the estimated depth maps and sparse ground truths.

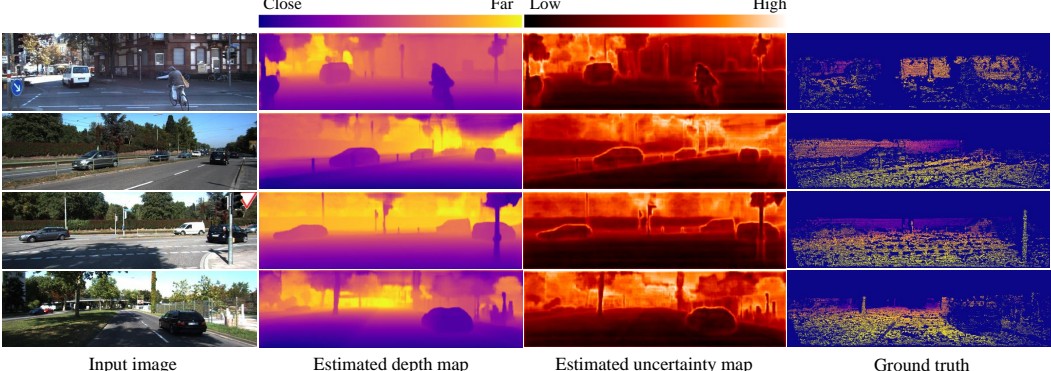

| Close | Far | Low | High |

| Input image | Estimated depth map | Estimated uncertainty map | Ground truth |

Figure 3: Qualitative results of our ADU-Depth with UEM. High uncertainty values are displayed in far background regions and object boundaries according to the estimated uncertainty maps.

## 4.4 Ablation Study

To better inspect the effect of each novel component in our distillation strategy, we conduct each module by an ablation study in Table 4. We also report the results produced by our teacher network in the first row as a reference. The simple yet effective design of teacher network achieves the promising performance by capturing additional 3D scene information in left-right image pairs.

Since the proposed distillation framework contains three main components, we conduct independent experiments with different combinations to further verify the effectiveness of each module. The attention-adapted feature distillation module shows a significant performance gain. It implies that the key to shrinking the monocular-stereo feature domain gap is to adaptively learn the teacher-student feature approximation. Then, UEM-guided distillation also achieves lower estimation errors due to learning for regions of high uncertainty and hard regions. Even though these two components have shown powerful effects, where the "Sq Rel" error is reduced from 0.188 to 0.154, the introduction of focal-depth adapted response distillation enables our ADU-Depth to achieve further accuracy boost and the optimal performance with the "Sq Rel" error 0.147. It benefits from the enhanced learning of hard-to-transfer knowledge from the teacher model.

## 5 Limitations

Although the proposed ADU-Depth can boost the performance of monocular depth estimation and learn 3D-aware knowledge from the left-right view based teacher model with novel distillation techniques (as shown in Table 4), our method cannot theoretically guarantee the applicability of depth estimation to 3D understanding. In future work, we will predict a volumetric scene representation and leverage novel view synthesis to provide spatial cues for monocular depth estimation.

## 6 Conclusion

We propose a novel distillation framework, ADU-Depth, to improve the performance of the monocular depth predictor, especially on distant objects as well as the silhouette of objects. To the best of our knowledge, we are the first supervised monocular depth estimation method that simulates the 3D-aware knowledge learned by a teacher network from left-right image pairs on a monocular student. To guarantee effective and smooth knowledge transfer to the monocular student, we design an attention-adapted feature imitation and a focal-depth based response imitation. In addition, we design an uncertainty estimation module to model the uncertainty of depth estimation, which guides the distillation in both feature space and result space. Extensive experimental results on the KITTI and DrivingStereo datasets show that our method achieves a new state-of-the-art performance.

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
