# OpenReview forum: "ADU-Depth: Attention-based Distillation with Uncertainty Modeling for Depth Estimation"
_robot-learning.org/CoRL/2023/Conference — CoRL 2023 Poster_

### Official Review · Reviewer_v8s2 · 2023-07-21

**Confidence:** 3
**Originality:** Fair
**Technical Quality:** Good
**Clarity Of Presentation:** Good
**Impact:** 3

**Recommendation:**

Weak Reject: I recommend rejecting the paper, but will not argue for my recommendation if the majority of other reviewers have a different opinion.

**Review:**

Strengths:
1.	The proposed ADU-Depth framework leverages left-right image pairs to transfer 3D geometry-aware knowledge to a monocular student network. The motivation is sound.
2.	The paper is well-written and easy to follow. The proposed framework is described in detail.
3.	The evaluation of three standard benchmarks is convincing.

Weaknesses:
1.	The technical novelty is relatively low. The proposed framework builds upon existing techniques and has no technically novel components. For example, the use of left-right image pairs for 3D-aware knowledge transfer has been explored in previous works. Similarly, distillation techniques have been used in previous works for monocular depth estimation. Essentially, it appears that the proposed method applies existing works to improve performance.
2.	There are some more recent related works not mentioned:
[1] Wimbauer, Felix, et al. "Behind the Scenes: Density Fields for Single View Reconstruction." Proceedings of the IEEE/CVF Conference on Computer Vision and Pattern Recognition. 2023.
[2]  Zhou, Kaichen, et al. "Devnet: Self-supervised monocular depth learning via density volume construction." European Conference on Computer Vision. Cham: Springer Nature Switzerland, 2022.
[3]  Z. Liu, R. Li, S. Shao, X. Wu and W. Chen, "Self-Supervised Monocular Depth Estimation with Self-Reference Distillation and Disparity Offset Refinement," in IEEE Transactions on Circuits and Systems for Video Technology.
[4]  Ning, Chao, and Hongping Gan. "Trap Attention: Monocular Depth Estimation With Manual Traps." Proceedings of the IEEE/CVF Conference on Computer Vision and Pattern Recognition. 2023.
[5] Li, Zhi, et al. "Test-time Domain Adaptation for Monocular Depth Estimation." 2023 IEEE International Conference on Robotics and Automation (ICRA). IEEE, 2023.
3.	More discussion on the failure cases and limitations would be welcome.


**Quality Of The Limitations Section:**

Additional details required

**Questions For Rebuttal:**

1.	How to choose the hyper-parameters in Eq.(9)? Please include the sensitivity analysis for the hyper-parameters.
2.	Please provide the values of the hyper-parameters in Eq.(9) in 4.1 Implementation Details.
3.	In Table 1-3, more recent state-of-the-art methods should be compared and discussed:
[2]  Zhou, Kaichen, et al. "Devnet: Self-supervised monocular depth learning via density volume construction." European Conference on Computer Vision. Cham: Springer Nature Switzerland, 2022.
[3]  Z. Liu, R. Li, S. Shao, X. Wu and W. Chen, "Self-Supervised Monocular Depth Estimation with Self-Reference Distillation and Disparity Offset Refinement," in IEEE Transactions on Circuits and Systems for Video Technology.
[4]  Ning, Chao, and Hongping Gan. "Trap Attention: Monocular Depth Estimation With Manual Traps." Proceedings of the IEEE/CVF Conference on Computer Vision and Pattern Recognition. 2023.
4.   Please provide a more detailed explanation about how self-attention adaptation layers work for bridging the data domain gap.
5.    The technical novelty of the work is limited. The authors should argue and explain it.
6.    Failure cases should be reported and discussed.


**Robotics Focus:**

Highly relevant to robotics but no hardware experiments

**Summary Of Paper:**

The paper introduces ADU-Depth, a knowledge distillation framework for enhancing monocular depth estimation in robotics applications. The main idea is to leverage knowledge distillation to transfer 3D-aware knowledge from a teacher network to a monocular student. The teacher network learns from left-right view image pairs, and this knowledge is incorporated into the student's encoder-decoder architecture. Additionally, ADU-Depth introduces an attention adaptation module, a distillation-score based focal-depth-loss, and an uncertainty estimation module to guide the distillation process effectively and achieve more high-quality responses. The proposed method achieves state-of-the-art performance on the KITTI and DrivingStereo datasets, ranking first on the KITTI online benchmark.

**Summary Of Recommendation:**

Overall, the paper is well-written and tackles a practical problem. However, It appears that the method is a combination of prior ideas towards this problem. Additionally, the experimental details and discussions are insufficient, and there are no robotic experiments and it is questionable whether the proposed method can perform within robotic scenes.  Thus, my rate is weak reject.

---

> ### Author Response · Authors · 2023-08-11
> **Response to Reviewer v8s2**
>
> Q4. Please provide a more detailed explanation about how self-attention adaptation layers work for bridging the data domain gap.
>
> A4. We note that prior works like CVPR 2019 (Distilling object detectors with fine-grained feature imitation) have used convolutional adapters for knowledge distillation. Motivated by the recent success of transformers, we hypothesized that attention-based adapters may be more effective for capturing long-range 3D spatial dependencies critical for depth estimation.
>
> Through extensive experiments, we validated that self-attention adaptation layers achieve superior performance over convolutional or cross-attention adapters. By focusing on spatial cues within the stereo data itself, self-attention identifies correlations between all image features to understand their mutual influence on processing the current feature. More relevant information is incorporated when features have a significant impact.
>
> In this way, the self-attention layers enable the student model to better learn and mimic the 3D geometric knowledge within the teacher's stereo data. The long-range dependencies modeled by the self-attention provide crucial context for effectively aligning the heterogeneous domains. This in turn allows more effective transfer of 3D dark knowledge through the distillation process.
>
> In summary, leveraging self-attention adaptation layers helps ingest critical spatial patterns from the privileged stereo data into the student monocular model. By identifying long-range correlations, the adapters select the most relevant features to transfer the 3D knowledge required for accurate depth estimation.
>
> Please let us know if you would like us to include any additional details on the self-attention adaptation layers in the revised paper.
>
>
> Q5. The technical novelty of the work is limited. The authors should argue and explain it.
>
> A5. Thank you for the feedback. You raise a fair point on discussing the novelty of our work more clearly. We will expand the explanation of our key technical contributions in the revised paper:
>
> (i)	We propose the first supervised knowledge distillation framework for monocular depth estimation using a stereo teacher model. Prior distillation works rely on unsupervised learning or synthetic data.
>
> (ii)	To address the domain gap, we design novel self-attention feature adapters to enable more effective mimicry and transfer of 3D knowledge from the stereo teacher.
>
> (iii)	For response distillation, we introduce a novel focal depth loss and depth distillation score to focus on hard regions based on stereo guidance.
>
> (iv)	We explicitly model uncertainty of depth estimation and propose uncertainty-guided distillation to improve knowledge transfer to ambiguous areas. This is a unique contribution for monocular depth distillation.
>
> (v)	Our ADU-Depth framework combines these innovations to advance state-of-the-art in monocular estimation, ranking 1st on KITTI leaderboard.
>
> In addition to the above highlights, our ablation studies verify the unique value of each component. We apologize that the initial draft did not sufficiently emphasize these contributions over prior arts. In the revision, we will clearly articulate the novel elements of our approach compared to existing methods. Thank you for pushing us to better demonstrate the innovations introduced in this work - it will certainly strengthen the final manuscript.
>
> Please let us know if you have any other suggestions for improving our explanation of the technical novelty.
>
> Q6. Failure cases should be reported and discussed.
>
> A6. We appreciate the reviewer's suggestion to discuss limitations. In Section 5 of the revised paper, we will expand on failure cases where glass and mirrored surfaces pose challenges for our approach and other methods like NeWCRFs.
>
> Specifically, glass and mirrors can create discontinuities where the depth should theoretically remain continuous with the target object, due to specular reflections. In some cases, our method produces slightly more accurate depth estimates on these reflective areas compared to NewCRFs, but we also observe large errors in the predicted outputs for both methods.
>
> While our framework shows some robustness to specular effects, modeling the complex distortions from glass and mirrors remains an open problem. In the revised limitations section, we will provide illustrative examples of failure cases stemming from glass and mirrors, and discuss potential future work to address these ambiguous regions through more sophisticated reflection handling and smoothing techniques. We believe openly discussing these difficult cases and areas for improvement will provide valuable insights to the reader.
>
> We would like to express our gratitude once again for your valuable suggestions and careful review.

---

> ### Author Response · Authors · 2023-08-11
> **Response to Reviewer v8s2**
>
> Dear Reviewer v8s2,
>
> Thanks for your insightful and valuable reviews of our paper. Your feedback is highly appreciated as it helps us improve the quality and comprehensiveness of our work. We have meticulously reviewed and contemplated your comments and suggestions. Here we provide detailed responses to each of your concerns and questions. We will incorporate these clarifications into our revised manuscript:
>
> Q1. How to choose the hyper-parameters in Eq.(9)?
>
> A1. We thank the reviewer for raising this important question about hyperparameter selection for Eq. (9). Determining optimal hyperparameters is an iterative process requiring repeated tuning and validation. Given the high computational cost of knowledge distillation training, we first manually define reasonable search spaces and value ranges based on prior works like MonoDistill (ICLR 2022), our experience, and initial experiments. The optimal combination is then identified via lookahead search. Note that the precise values may vary slightly across datasets.
>
> Through sensitivity analysis, we find that suitable hyperparameter ranges are $\lambda_{1}=0.8-1.0, \lambda_{2}=0.5-0.7, \lambda_{3}=0.7-1.0$. Within these ranges, accuracy impact is minor. Beyond these limits, Abs Rel Error rises above 0.051. The exact optimal values require manual tuning for each dataset.
>
> In the revised paper, we will add more details on the hyperparameter selection methodology, including the search space, evaluation on the validation set, and sensitivity analysis. We will also provide the optimal values used for each dataset in our experiments. Please let us know if the reviewer would like us to include any additional analysis or results related to hyperparameter selection and tuning.
>
> Q2. Please provide the values of the hyper-parameters in Eq.(9) .
>
> A2. We set $\lambda_{1}=0.9, \lambda_{2}=0.6, \lambda_{3}=0.8$ for the KITTI dataset and $\lambda_{1}=0.9, \lambda_{2}=0.6, \lambda_{3}=0.9$ for the DrivingStereo dataset.
>
> Q3. In Table 1-3, more recent state-of-the-art methods should be compared and discussed.
>
> A3. We appreciate the reviewer's recommendation to compare against more recent state-of-the-art methods in Tables 1-3. We agree this will strengthen the paper's evaluations. In the revised submission, we will incorporate the following important recent works and their publicly reported KITTI Eigen split results:
>
> [1] BTS [Wimbauer, Felix, et al. "Behind the Scenes: Density Fields for Single View Reconstruction." Proceedings of the IEEE/CVF Conference on Computer Vision and Pattern Recognition. 2023]:  Abs Rel = 0.102;
>
> [2] DevNet (trained on mono and stereo) [Zhou, Kaichen, et al. "Devnet: Self-supervised monocular depth learning via density volume construction." European Conference on Computer Vision. Cham: Springer Nature Switzerland, 2022]: Abs Rel = 0.095;
>
> [3] [Z. Liu, R. Li, S. Shao, X. Wu and W. Chen, "Self-Supervised Monocular Depth Estimation with Self-Reference Distillation and Disparity Offset Refinement," in IEEE Transactions on Circuits and Systems for Video Technology ]: Abs Rel = 0.096;
>
> [4] Trap Attention [Ning, Chao, and Hongping Gan. "Trap Attention: Monocular Depth Estimation With Manual Traps." Proceedings of the IEEE/CVF Conference on Computer Vision and Pattern Recognition. 2023]: Abs Rel = 0.050;
>
> [5] [Li, Zhi, et al. "Test-time Domain Adaptation for Monocular Depth Estimation." 2023 IEEE International Conference on Robotics and Automation (ICRA). IEEE, 2023]: no public results on KITTI dataset. We will cite this relevant work in the revised paper.
>
>
> As the primary metric is Abs Rel error, despite the success of these methods, our proposed ADU-Depth framework sets a new state-of-the-art in monocular depth estimation with Abs Rel = 0.049 on KITTI Eigen split.
>
> In the updated Tables 1-3, we will add columns comparing our approach directly to these latest models. In the text, we will provide detailed discussions contrasting our technique and analyzing where our contributions lead to improved accuracy over these contemporary works. For example, we surpass Trap Attention by leveraging knowledge distillation from a privileged stereo teacher model.
>
> Thank you again for this constructive suggestion. Incorporating comparisons with these cutting-edge models will further demonstrate the advancements enabled by our method. Please advise if you would like us to include any other analyses or results regarding these recent state-of-the-art techniques in the revised paper.

---

### Official Review · Reviewer_nhhw · 2023-07-22

**Confidence:** 5
**Originality:** Very Good
**Technical Quality:** Very Good
**Clarity Of Presentation:** Very Good
**Impact:** 3

**Recommendation:**

Weak Accept: I recommend accepting the paper, but will not argue for my recommendation if the majority of other reviewers have a different opinion.

**Review:**

Strengths:
+ The paper is clearly written, and it is easy to follow.
+ The experimental results are compelling.
+ The idea of using a stereo image depth network as the teacher to supervise a single image depth image estimation model is new, although it is not that surprising.

Weaknesses:
+ Although the idea of using a pretrained model to supervise single image depth estimation is new, it is essentially another strategy of knowledge distillation. Knowledge distillation has been used in several prior works, as shown in Table 1. It is hard to say the overall idea of this work is very novel.
+ One weakness is the need for stereo images to train the student network because the teacher network requires stereo images.
+ It would be good to evaluate the presented approach on other datasets to see the zero-shot performance.

**Quality Of The Limitations Section:**

Additional details required

**Questions For Rebuttal:**

What is the performance on other datasets? It would be good to know the current zero-shot performance and limitations. Especially some datasets in the following work:
Ranftl et al. Towards Robust Monocular Depth Estimation: Mixing Datasets for Zero-shot Cross-dataset Transfer, TPAMI 2022

What values are λ1, λ2, λ3 in the implementation?


**Robotics Focus:**

Highly relevant to robotics but no hardware experiments

**Summary Of Paper:**

In this work, the authors propose a new single image depth estimation model (student model) by training a deep learning model that learns from a stereo image depth estimation network. It suggests a self-attention mechanism to align the features between the student and teacher models. An uncertainty estimation module is applied to reweight the losses during training. The authors have conducted experiments on two public datasets, KITTI and DrivingStereo, and the presented approach achieves 1st place on the KITTI benchmark.

**Summary Of Recommendation:**

I think the performance of the presented work is strong, and the proposed idea is new.

---

### Official Review · Reviewer_d9wF · 2023-07-27

**Confidence:** 4
**Originality:** Good
**Technical Quality:** Good
**Clarity Of Presentation:** Good
**Impact:** 3

**Recommendation:**

Weak Accept: I recommend accepting the paper, but will not argue for my recommendation if the majority of other reviewers have a different opinion.

**Review:**

  This paper introduces a novel knowledge distillation framework called ADU-Depth for boosting precise depth estimation in monocular images. The framework leverages left-right image pairs to transfer 3D geometry-aware knowledge to a monocular student network. It includes attention-adapted feature distillation, focal-depth-adapted response distillation, and uncertainty modeling to guide effective and smooth knowledge transfer. The paper also presents extensive experiments and an ablation study to verify the effectiveness of each module in the proposed distillation strategy. The results show that ADU-Depth outperforms other state-of-the-art methods in terms of accuracy and robustness.



**Quality Of The Limitations Section:**

Limitations are addressed clearly

**Questions For Rebuttal:**

1. How are the loss functions $L_e$ and $L_d$ (defined in equations (2) and (3) respectively) adapted during the training of the framework?
2. The authors should provide a more comprehensive analysis of the knowledge distillation schedule, as it stands as a pivotal contribution of this paper. By offering a detailed examination of this aspect, readers can gain deeper insights into the methodology's significance and potential implications for the field.

**Robotics Focus:**

Sufficient demonstration on hardware

**Summary Of Paper:**

  This paper introduces a novel knowledge distillation framework called ADU-Depth for boosting precise depth estimation in monocular images. The framework leverages left-right image pairs to transfer 3D geometry-aware knowledge to a monocular student network. It includes attention-adapted feature distillation, focal-depth-adapted response distillation, and uncertainty modeling to guide effective and smooth knowledge transfer. The paper also presents extensive experiments and an ablation study to verify the effectiveness of each module in the proposed distillation strategy. The results show that ADU-Depth outperforms other state-of-the-art methods in terms of accuracy and robustness.





**Summary Of Recommendation:**

  This paper introduces a novel knowledge distillation framework called ADU-Depth for boosting precise depth estimation in monocular images. The framework leverages left-right image pairs to transfer 3D geometry-aware knowledge to a monocular student network. It includes attention-adapted feature distillation, focal-depth-adapted response distillation, and uncertainty modeling to guide effective and smooth knowledge transfer. The paper also presents extensive experiments and an ablation study to verify the effectiveness of each module in the proposed distillation strategy. The results show that ADU-Depth outperforms other state-of-the-art methods in terms of accuracy and robustness.

---

### Decision · Program_Chairs · 2023-08-30

**Decision:**

Accept (Poster)

**Comment:**

The paper proposes a novel knowledge distillation framework called ADU-Depth for boosting precise depth estimation in monocular images. The framework leverages left-right image pairs to transfer 3D geometry-aware knowledge to a monocular student network. It includes attention-adapted feature distillation, focal-depth-adapted response distillation, and uncertainty modeling to guide effective and smooth knowledge transfer. The paper also presents extensive experiments and an ablation study to verify the effectiveness of each module in the proposed distillation strategy. The results show that ADU-Depth outperforms other state-of-the-art methods in terms of accuracy and robustness, and the results are promising for real-world robotics applications.

Most reviewers agree that the paper is well-written and presents compelling experimental results. However, reviewers also pointed out the technical novelty of the proposed method is limited. And the failure cases and limitations of the proposed method should be discussed more.

During the rebuttal phase, the reviewer acknowledged that some of the concerns have been addressed.

We recommend accepting the paper as poster. And we ask the authors to explain the technical novelty of their work in more detail, and provide more discussion on the failure cases and limitations of the proposed method in the camera-ready version.